# Potential Transient Response of Terrestrial Vegetation and Carbon in Northern North America from Climate Change

**Steven A. Flanagan** [1,2,*], **George C. Hurtt** [1], **Justin P. Fisk** [3], **Ritvik Sahajpal** [1], **Maosheng Zhao** [4], **Ralph Dubayah** [1], **Matthew C. Hansen** [1], **Joe H. Sullivan** [5] and **G. James Collatz** [4]

1   Department of Geographical Sciences, University of Maryland, College Park, MD 20740, USA; gchurtt@umd.edu (G.C.H.) ; ritvik@umd.edu (R.S.); dubayah@umd.edu (R.D.); mhansen@umd.edu (M.C.H.)
2   Wildland Fire Science Program, Tall Timbers Research Station, 13093 Henry Beadel Drive, Tallahassee, FL 32312, USA
3   Applied GeoSolutions, 87 Packers Falls Road, Durham, NH 03824, USA; jfisk@appliedgeosolutions.com
4   NASA Goddard Space Flight Center, Greenbelt, MD 20771, USA; maosheng.zhao@nasa.gov (M.Z.); george.j.collatz@nasa.gov (G.J.C.)
5   Department of Plant Science and Landscape Architecture, University of Maryland, College Park, MD 20740, USA; jsull@umd.edu
*   Correspondence: sflanaga@umd.edu; Tel.: +1-914-262-9221

**Abstract:** Terrestrial ecosystems and their vegetation are linked to climate. With the potential of accelerated climate change from anthropogenic forcing, there is a need to further evaluate the transient response of ecosystems, their vegetation, and their influence on the carbon balance, to this change. The equilibrium response of ecosystems to climate change has been estimated in previous studies in global domains. However, research on the transient response of terrestrial vegetation to climate change is often limited to domains at the sub-continent scale. Estimation of the transient response of vegetation requires the use of mechanistic models to predict the consequences of competition, dispersal, landscape heterogeneity, disturbance, and other factors, where it becomes computationally prohibitive at scales larger than sub-continental. Here, we used a pseudo-spatial ecosystem model with a vegetation migration sub-model that reduced computational intensity and predicted the transient response of vegetation and carbon to climate change in northern North America. The ecosystem model was first run with a current climatology at half-degree resolution for 1000 years to establish current vegetation and carbon distribution. From that distribution, climate was changed to a future climatology and the ecosystem model run for an additional 2000 simulation years. A model experimental design with different combinations of vegetation dispersal rates, dispersal modes, and disturbance rates produced 18 potential change scenarios. Results indicated that potential redistribution of terrestrial vegetation from climate change was strongly impacted by dispersal rates, moderately affected by disturbance rates, and marginally impacted by dispersal mode. For carbon, the sensitivities were opposite. A potential transient net carbon sink greater than that predicted by the equilibrium response was estimated on time scales of decades–centuries, but diminished over longer time scales. Continued research should further explore the interactions between competition, dispersal, and disturbance, particularly in regards to vegetation redistribution.

**Keywords:** climate change; earth system modeling; ecosystem demography model; migration; plant ecology; plant migration; transient response



## 1. Introduction

Forests contain ~80% of above ground carbon and sequester ~30% of annual fossil fuel emissions, and thus have a prominent role in the carbon balance [1,2]. The distribution of terrestrial ecosystems is strongly influenced by climate [3–6], so how ecosystems reorganize from climate change presents an important research area in regards to terrestrial carbon. Paleoecological records predicted forest migration rates during the last glacial period greater than considered possible [7–11]. Two theories for how this happened are rapid migration and refugial populations, and is known as Reid's Paradox [12–14]. For large domain studies that predict the potential redistribution of vegetation by plant migration due to expected future climate change, Dynamic Global Vegetation Models (DGVMs) or Earth System Models (ESMs) are used. Research at these domain sizes often implement scaling strategies at the cost of some fine scale processes, such as individual-based plant migration, to reduce computational requirements. Therefore, improvements to their underlying vegetation demographics are continued and are important research topics, especially when predicting the redistribution of vegetation from plant migration due to climate change.

Most DGVMs are cohort, not individual, based, and given the complexities of dispersal between grid cells, they approximate the transient response of plant migration due to climate change through other methods [15,16]. TRIFFID [17] leaves a fraction of its seed bank in all cells, so if climate changes, better adapted species may alter the species composition. Sheffield-DGVM (SHE) [18,19], ORCHIDEE [20], and Lund-Postdam-Jena (LPJ) [21] have establishment of climatically favored plant functional types (PFTs). However, there is no between grid cell dispersal due to the complexity of this interaction, and partially because they are not individually based. Two DGVMs are individually based, and have attempted to simulate the transient response of vegetation between grid cells. SEIB-DGVM [22,23] is individually based and simulated migration between cells in Africa. For each half-degree cell, a 30 m × 30 m forest gap model was run and unlimited vs. no-migration simulated. The difference in the size of the cell and the spatial extent of the gap model did not allow for simulations at a specific dispersal distance. LPJ-GUESS [24,25] had dispersal between patches (smaller areas within a grid cell), and calculated the probability of spread to a neighboring cell based on a dispersal kernel, but only in an idealized landscape. The difficulties of simulating this fine scale process in a larger domain means most dispersal and migration studies continue to occur in Forest Landscape Models such as TreeMig [26], which has simulated regions of European Forest, and LANDIS-PRO [27,28], which has simulated multiple regions of the US. Moving beyond regional scales of simulating the transient response of plant migration with explicit dispersal remains a challenge.

Another DGVM, the Ecosystem Demography (ED) [29,30], is individually based. One of the reasons it is able to simulate large domains is that it is pseudo-spatial. Within every grid cell, the number of individuals of a PFT and their size and age, and the area they occupy (patch size) are known, but the explicit location in the grid cell is not. Recently, a pseudo-spatial dispersal sub-model was developed and implemented in ED [31]. It simulates the spatially explicit process of plant migration from dispersal in the pseudo-spatial framework of ED. Here, previous research where the PFT distribution in ED for northern North America was validated with remote sensing data, and then a climate change scenario run that showed the equilibrium response of vegetation and carbon was used with the new dispersal sub-model, and the transient response from individual-based dispersal in large domains explored with multiple scenarios. The migration sub-model is run with a model experimental designed to investigate the impact of (1) dispersal distance, (2) dispersal mode, and (3) disturbance rate on the potential transient redistribution of terrestrial vegetation and carbon from climate change in northern North America over a range of time scales (years–millennia).

## 2. Materials and Methods

### 2.1. Model

ED [29,30] is a mechanistic model that uses a size and age-structured approximation for the first moment of the spatial stochastic process of vegetation dynamics. The size and age-structured approximation means it is an individual-based model of vegetation dynamics that is pseudo-spatial instead of spatially explicit. Individuals compete mechanistically for water, nutrients, and light, governed by sub-models of growth, mortality, water, phenology, biodiversity, disturbance, hydrology, and soil biogeochemistry. Plants in ED are represented by PFTs, which group vegetation into classes dependent on physiognomy, leaf form, photosynthetic pathway, and other characteristics, and are adjusted for the region of study. Following Hurtt et al. [32], trees in North America are represented by two dominant types, cold deciduous and evergreen, with the modifications made by Flanagan et al. [33]. That research used advanced remote sensing to calibrate ED for the proper PFT distribution under contemporary climate, and then simulated a climate change scenario to determine the equilibrium response. Those findings supported additional research into the transient response of plant migration in that domain under that climate scenario. ED has also been successfully implemented in South, Central, and North America, as well as Mozambique [34–40]. It is currently being used in NASA's Carbon Monitoring System (CMS) [39,41], and the NASA Global Ecosystem Dynamics Investigation (GEDI) mission [42].

The dispersal sub-model in ED [31] replicates the spatially explicit process of dispersal in its pseudo-spatial environment. Though pseudo-spatial, ED is still individually based. Inside of a grid cell, seeds are produced by individuals in a known area called a patch, that is a fraction of the grid cell. With a given dispersal distance, the proportion of seeds produced by a species expected to disperse outside the patch is calculated by the relationship between the size of the patch and the dispersal distance. Then, the size of the patch is related to the grid cell size to determine the final proportion of seeds that would enter a new grid cell. Between grid cell dispersal is a function of a chosen dispersal distance, the seeds produced by individuals and the size of the patch they were produced on, and the size of the grid cell.

A model experimental design that used the dispersal sub-model evaluated the impact of dispersal rate, dispersal mode, and disturbance rate on the transient response of migration to climate change in northern North America (40°N to 75°N, and 165°W to 50°W). Eighteen cases were considered; dispersal rates of 0.1 km, 1 km, and 10 km; disturbance rates at the model's standard 1.2% per year, doubled and tripled; and directed or even dispersal. The maximum migration rate for many species in this area is 1 km per year [43], so that was the median dispersal rate, and then an order of magnitude higher and lower were chosen. The disturbance rates were selected based on Dolan et al. [36], who used a similar region with the same model and climate scenario to examine climate change and disturbance impacts on these forests. To better examine competition, dispersal was "directed" to an area that was recently disturbed, and hence would have limited competition for light and nutrients or "even"ly spread proportionally across the entire grid cell. Each scenario started with a 1000 years run of the current climatology to stabilize the initial biomass and PFT distribution. Exploration of initial results informed the decision to simulate an additional 2000 years with the future climatology. The climate was abruptly changed as the thirty-year gap between climatologies was deemed minor with the length of the simulations. The plant migration transient response results were evaluated by percent total carbon and percent dominant PFT type of the equilibrium future climate distribution they predicted at the end of the simulation and through time. Dominant PFT of a grid cell was determined by applying the National Land Cover Dataset 1992 (NLCD92) [44] classification for forest composition of 75% cover of a particular type, deciduous or evergreen, otherwise the forest was classified as mixed. Sites below 25% cover were considered non-forest.

## 2.2. Climate Data

Two climate data sets were used. A current climate data set established contemporary carbon and PFT distribution as supported by remote sensing data [33], and a future climate data set established the predicted equilibrium responses and was used for the transient response scenarios. The current climate data set was from the Multi-Scale Synthesis and Terrestrial Model Intercomparison Project (MsTMIP) conducted by the North America Carbon Program (NACP) [45,46]. It is a combination of the Climate Research Unit (CRU) and National Centers for Environmental Prediction (NCEP) climatologies at $0.5 \times 0.5$ degree global resolution from 1901–2010 in a WGS84 projection at 6 hourly time steps. The future climate data set was from the North American Climate Change Assessment Program (NARCCAP), which produces multiple future climatologies at ~50 km resolution [47]. The choice to use this data was made because ED requires specific humidity inputs, and this program is one of the few which includes this in the climate change scenarios. Future climate projections are provided by coupling a set of regional climate models (RCMs) driven by a set of atmosphere-ocean general circulation models (AOGCMs) forced with the Special Report on Emission Scenarios (SRES) A2 scenario for the 21st century. The A2 scenario is the only one currently simulated. One of the first combinations available to the community, and hence used, was the Community Climate System Model (CCSM) as the driving model and MM5I as the regional model, and contained future climate data from 2041–2070 at 3 hourly time steps in a Lambert Conic Conformal projection. The climate is warmer and wetter. The NARCCAP climate data set was converted to half-degree resolution with a WGS84 projection to match the current climate data set.

## 3. Results

### 3.1. Comparison of the Transient Response of Migration to the Equilibrium Response

The 18 scenarios were compared to the equilibrium response (what the model predicted the distribution would be under the future climatology and no dispersal) that represented the disturbance rate they were run at. As disturbance alters final carbon stocks and vegetation distribution, three equilibrium response scenarios served as controls for six transient response scenarios each (Figure 1). Figure 1A,B respectively show the predicted current PFT distribution calibrated to match remote sensing data and the equilibrium response distribution to a climate change scenario as shown in Flanagan et al. [33]. Figure 1C,D are what the equilibrium response to the climate change scenario is under doubled and tripled disturbance rates.

To aid in the identification of factors that influenced carbon and vegetation redistribution, the final results (simulation year 2000) from transient migration scenarios were compared with their corresponding equilibrium cases. Comparison of total carbon ranged from 94% to 116% with a mean value of 107%, and PFT sites matched ranged and 60% to 86% with a mean of 74% (Figure 2 and Table 1).

To further isolate the magnitude of the effects, the average Root Mean Square Error (RMSE) (% units) for each independent variable between the responses were calculated to be used as a descriptive value only. PFT redistribution was strongly impacted by dispersal rates, moderately affected by disturbance rates, and marginally impacted by dispersal mode. The sensitivities were opposite for carbon. The magnitudes on PFT distribution were; dispersal rate 14.1 ± 4.8%, disturbance rate 7.8 ± 2.7%, and dispersal mode 3.7 ± 2.3% For total carbon the magnitudes were; dispersal mode 9.0 ± 4.3%, disturbance rate 7.5 ± 2.2%, and dispersal rate 6.5 ± 3.2% (Figure 3).

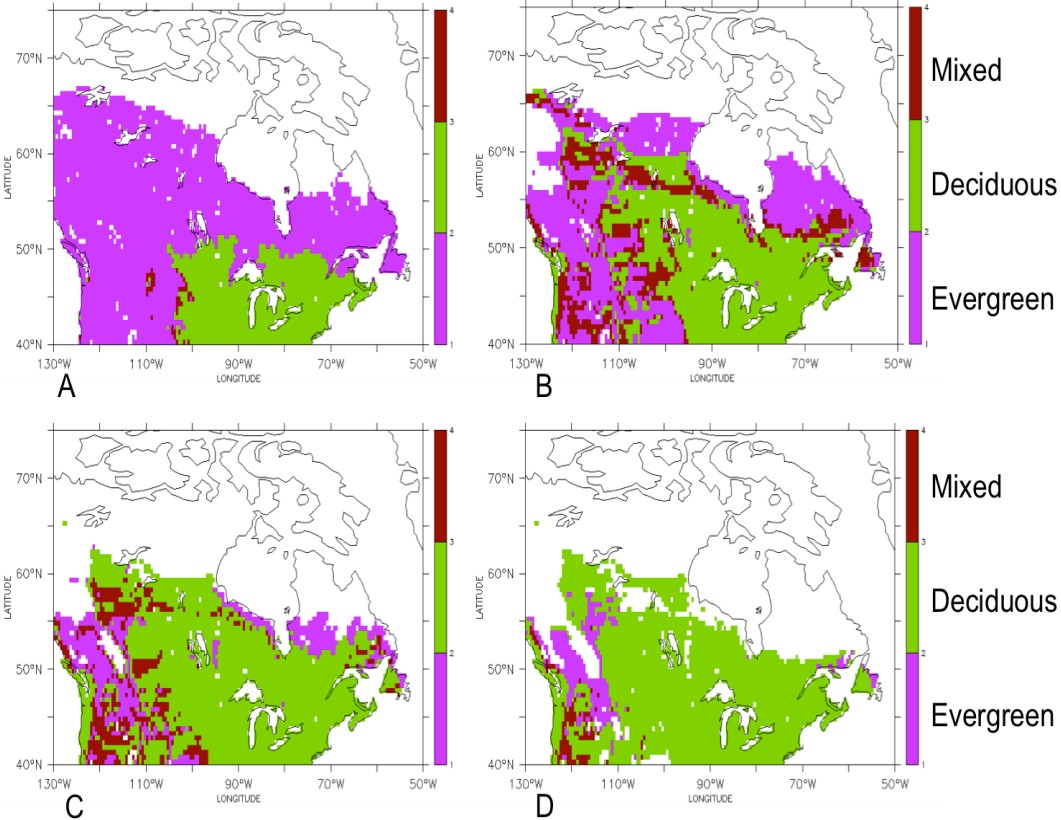

**Figure 1.** The dominant plant functional types (PFT) distribution (**A**) with current climate and the equilibrium response distribution for (**B**) future climate, (**C**) future climate with disturbance rate doubled, and (**D**) future climate with disturbance rate tripled.

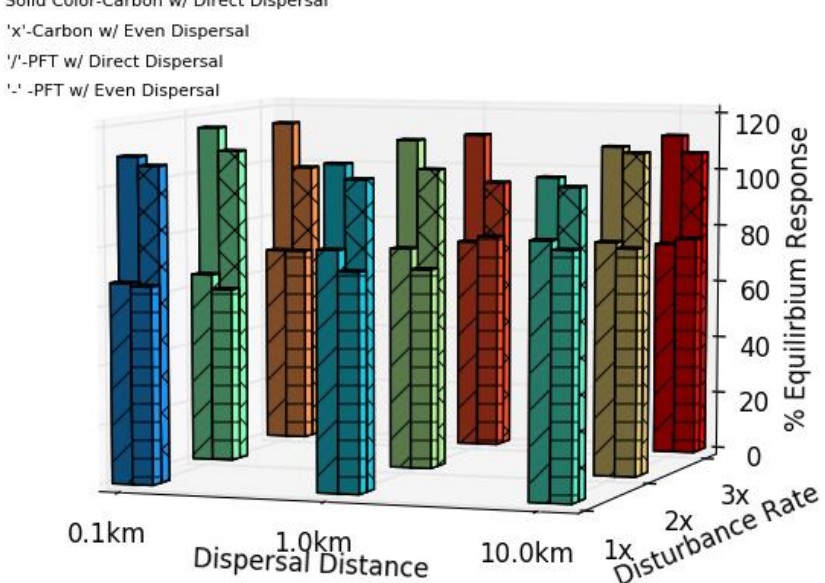

**Figure 2.** The percentage of the equilibrium responses total carbon and the percentage of dominant PFT sites matched that each scenario had obtained by the final simulation year.

**Table 1.** The listed percentage of the equilibrium responses total carbon and the percentage of dominant PFT sites matched that each scenario had obtained by the final simulation year.

| Scenario | Percentage of Equilibrium Response Carbon | | Percentage of Equilibrium Response Dominant PFT Sites | |
|---|---|---|---|---|
| | Direct | Even | Direct | Even |
| 0.1 km distance 1x disturbance | 107 | 110 | 68 | 67 |
| 1.0 km distance 1x disturbance | 104 | 109 | 81 | 74 |
| 10.0 km distance 1x distrubance | 103 | 106 | 86 | 83 |
| 0.1 km distance 2x disturbance | 108 | 116 | 65 | 60 |
| 1.0 km distance 2x disturbance | 103 | 113 | 76 | 69 |
| 10.0 km distance 2x distrubance | 110 | 112 | 80 | 78 |
| 0.1 km distance 3x disturbance | 98 | 114 | 68 | 68 |
| 1.0 km distance 3x disturbance | 94 | 111 | 73 | 75 |
| 10.0 km distance 3x distrubance | 106 | 112 | 74 | 76 |

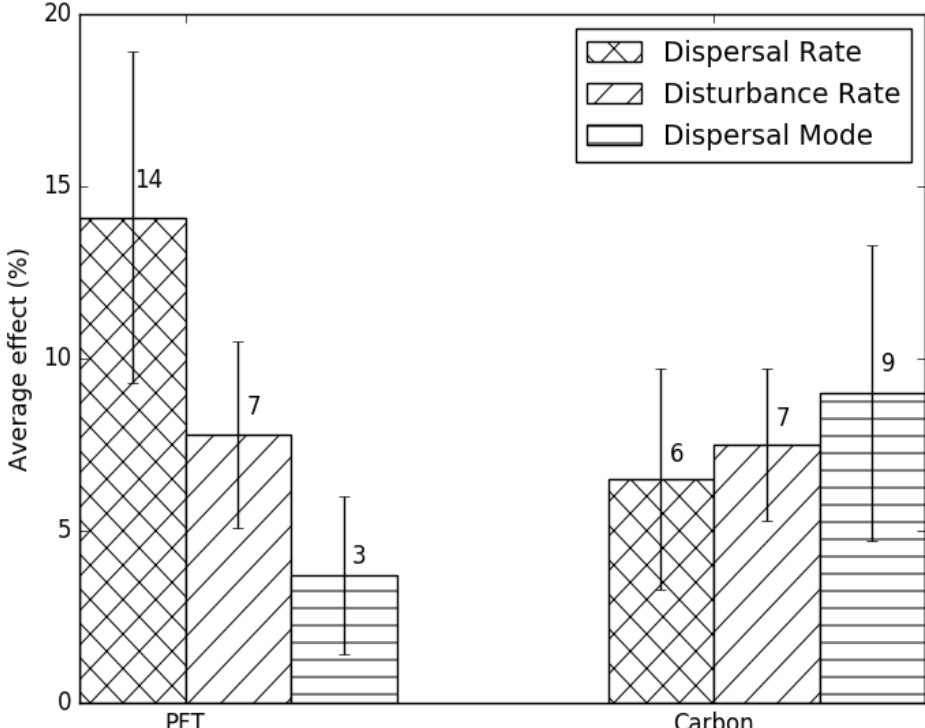

**Figure 3.** The Root Mean Square Error (RMSE) (% units) of the average effect that each independent variable had on the percentage of the equilibrium responses total carbon and the percentage of dominant PFT sites matched that each transient response scenarios had achieved by the final simulation year.

The transient response of plant migration increased the northern limit of the forest as sites that would not be classified as forest (<2 $Kg/m^2$ AGB) now met that threshold from migration. This phenomenon was responsible for the only two cases where the percentage of equilibrium carbon predicted by the transient response was less than 100 percent. A higher disturbance rate decreased overall total carbon, and with migration a number of new sites contained biomass, but did not meet the classification for forest (Figure 4). Total carbon increased in all transient scenarios but was not always captured based on the definition of a forest used as migration redistributed the concentration of biomass.

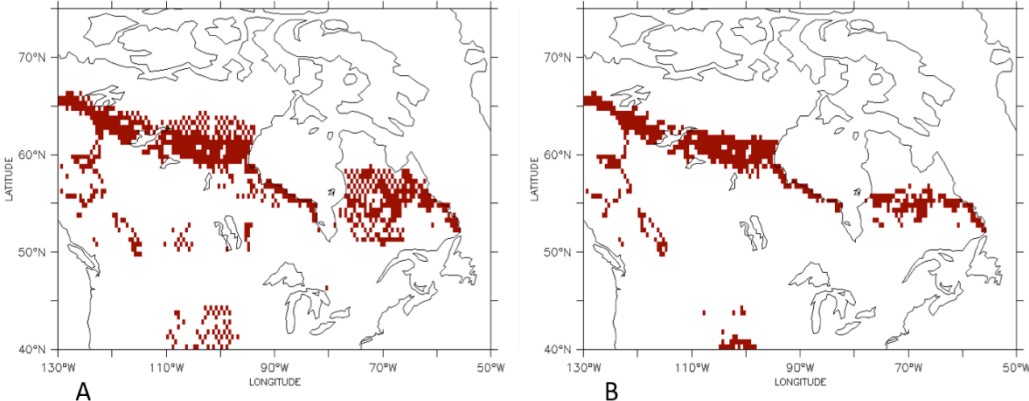

**Figure 4.** Maps of sites that contain biomass below the threshold of forest classification (in red) for directed (**A**) and even (**B**) dispersal at 0.1 km per year with a tripled disturbance rate.

Dispersal rate had the largest effect on vegetation redistribution. As an example, when the 0.1 km and 10 km dispersal rates of directed dispersal at the standard disturbance are compared to the equilibrium scenario, most sites matched with the 10 km rate and the northern extent of the forest had increased while the 0.1 km rate was still migrating across the landscape (Figure 5).

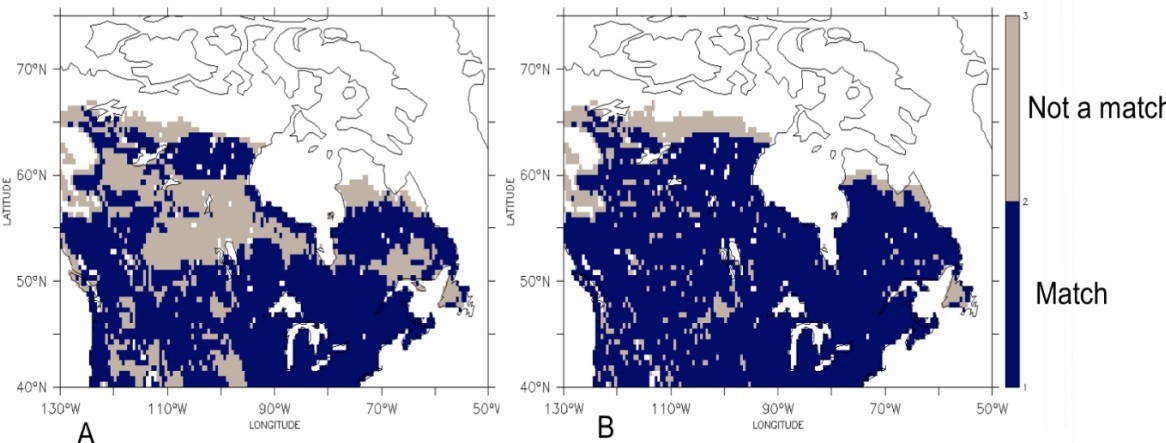

**Figure 5.** Map comparisons of predicted dominant PFT distribution in the final simulation year for directed dispersal and standard disturbance with a dispersal rate of (**A**) 0.1 km and (**B**) 10 km to the equilibrium response.

The case of directed dispersal with 0.1 km dispersal distance and standard disturbance rate matched 68% of the predicted equilibrium response sites at the end of the simulation (Figure 5A). The same scenario with 10 km dispersal distance matched 86% of the sites (Figure 5B). None of the transient plant migration scenarios reached 100% of the equilibrium dominant PFT pattern due to a combination of different effects. Limited dispersal distance prevented migration to the farthest predicted equilibrium locations (Figure 5A, gray in middle and west). High dispersal rates matched the majority of the predicted equilibrium response at the end of the simulation (Figure 5B, blue) but in all cases migration maintained forest area farther north than the equilibrium scenario predicted, as the plants migrating increased biomass above the forest classification threshold (Figure 5A,B, gray at top). These factors prevented the percentage of sites where the PFT distribution matched that of the equilibrium response from approaching 100% in all scenarios (Figure 2, Table 1). Comparison of the total difference in carbon to the portions coming from the evergreen PFT highlighted this effect (Figure 6).

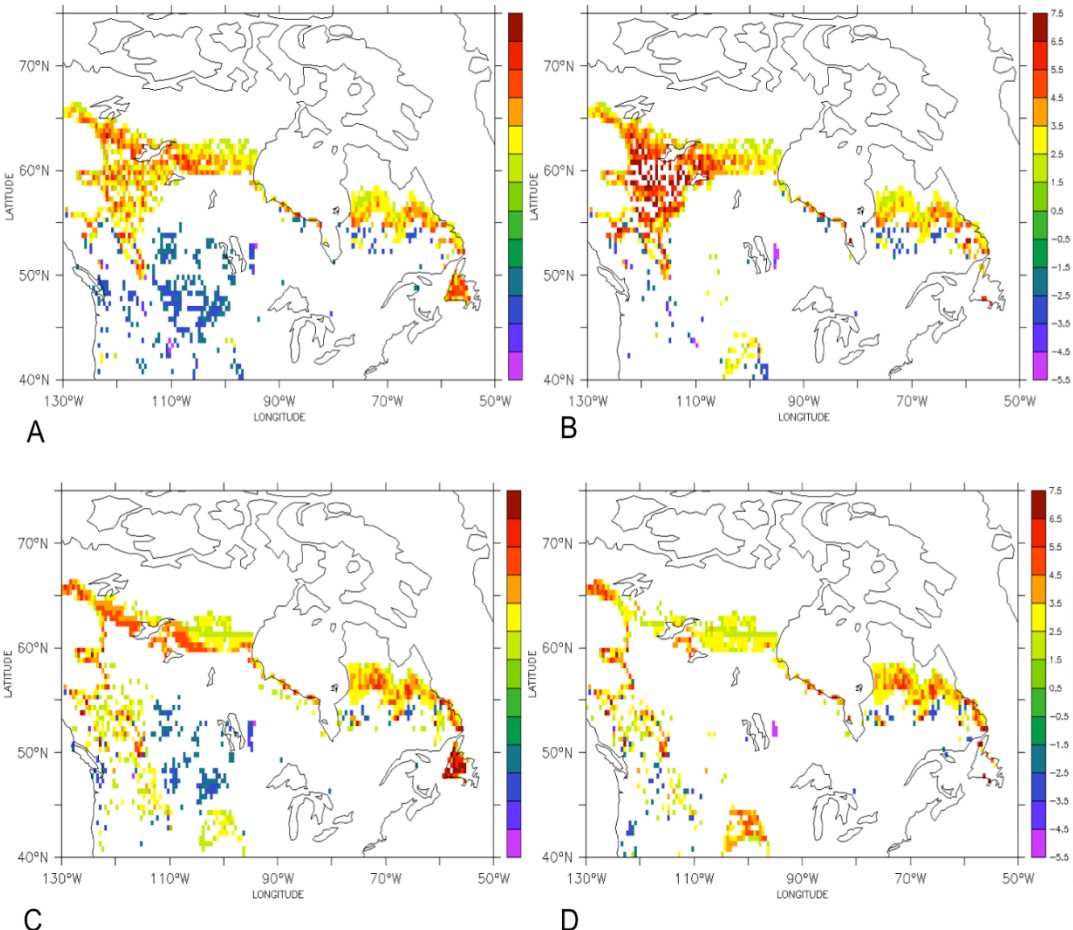

**Figure 6.** Map of the areas where the total biomass at the end of the transient simulation differs from that of the equilibrium simulation by ±2 Kg/m$^2$ for directed dispersal at (**A**) 1 km and (**C**) 10 km per year and the corresponding amounts, (**B**) and (**D**), that come from evergreen biomass. The deciduous PFT had not finished its expected migration in the 1 km simulation, so it still has a larger evergreen component (B increased biomass in the NW). Both simulations increased the northern extent of the forest as evergreen species migrated.

Disturbance rate had an intermediate effect on both carbon and vegetation redistribution. Increased disturbance decreased the time it took the deciduous PFT to migrate (Figure 7), but lowered total carbon.

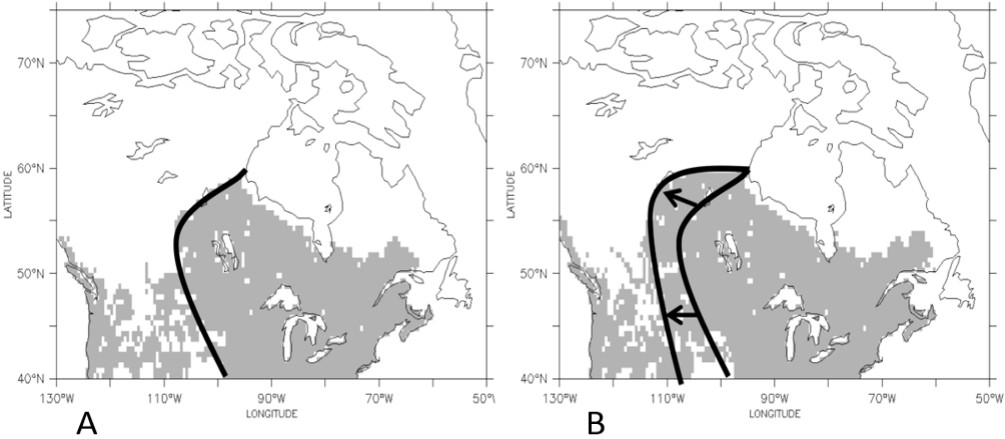

**Figure 7.** Map of the extent that deciduous forest migrated with directed dispersal at (**A**) 1 km a year with the standard disturbance rate, and (**B**) double the disturbance rate.

## 3.2. Temporal Patterns of Migration

The temporal response of plant migration showed that the evergreen PFT increased its carbon stocks after the start of the simulation. This caused total carbon stocks to overshoot the predicted future equilibrium total carbon before approaching it as migration occurred (Figure 8). All scenarios demonstrated this same trend but differed in the time it took to approach equilibrium.

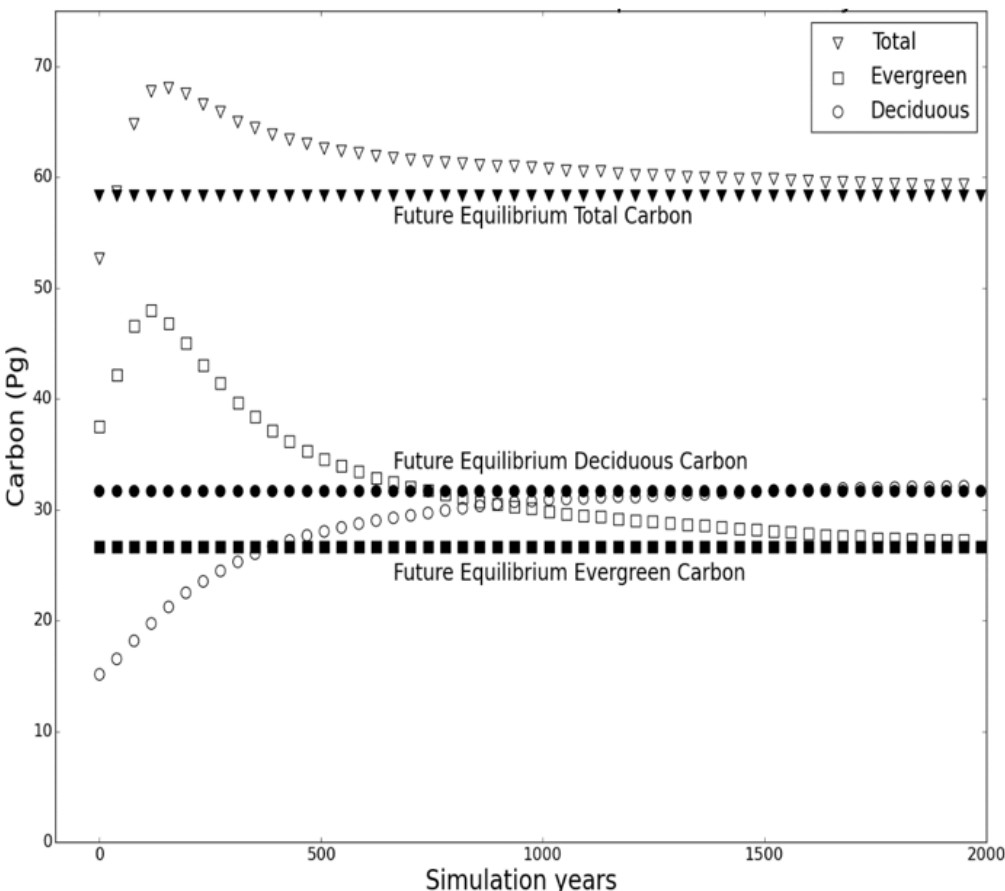

**Figure 8.** The temporal response from plant migration of carbon and the contribution from each PFT for the case of directed dispersal at 10 km per year under standard disturbance. The predicted equilibrium response total carbon, and the portion from each PFT, are shown in black. The transient response total carbon and the portion from each PFT are shown in white and change with simulation year.

The transient response from migration produced potential carbon storage that exceeded that of the equilibrium response because of the time it took for the new dominant PFT to migrate. Though deciduous species eventually established in areas that were initially evergreen, until that occurred, the evergreen species stored slightly more total carbon than the deciduous species did once it finished migrating (Figure 9). After 100 simulation years, the difference in total carbon from the equilibrium response showed an increase in carbon at many sites (Figure 9A, green). The difference from the evergreen PFT showed it was responsible for the increased carbon (Figure 9B). At the end of the simulation, carbon typically matched the equilibrium response (Figure 9C), except where evergreen migration had expanded the northern extent of the forest (Figure 9C,D).

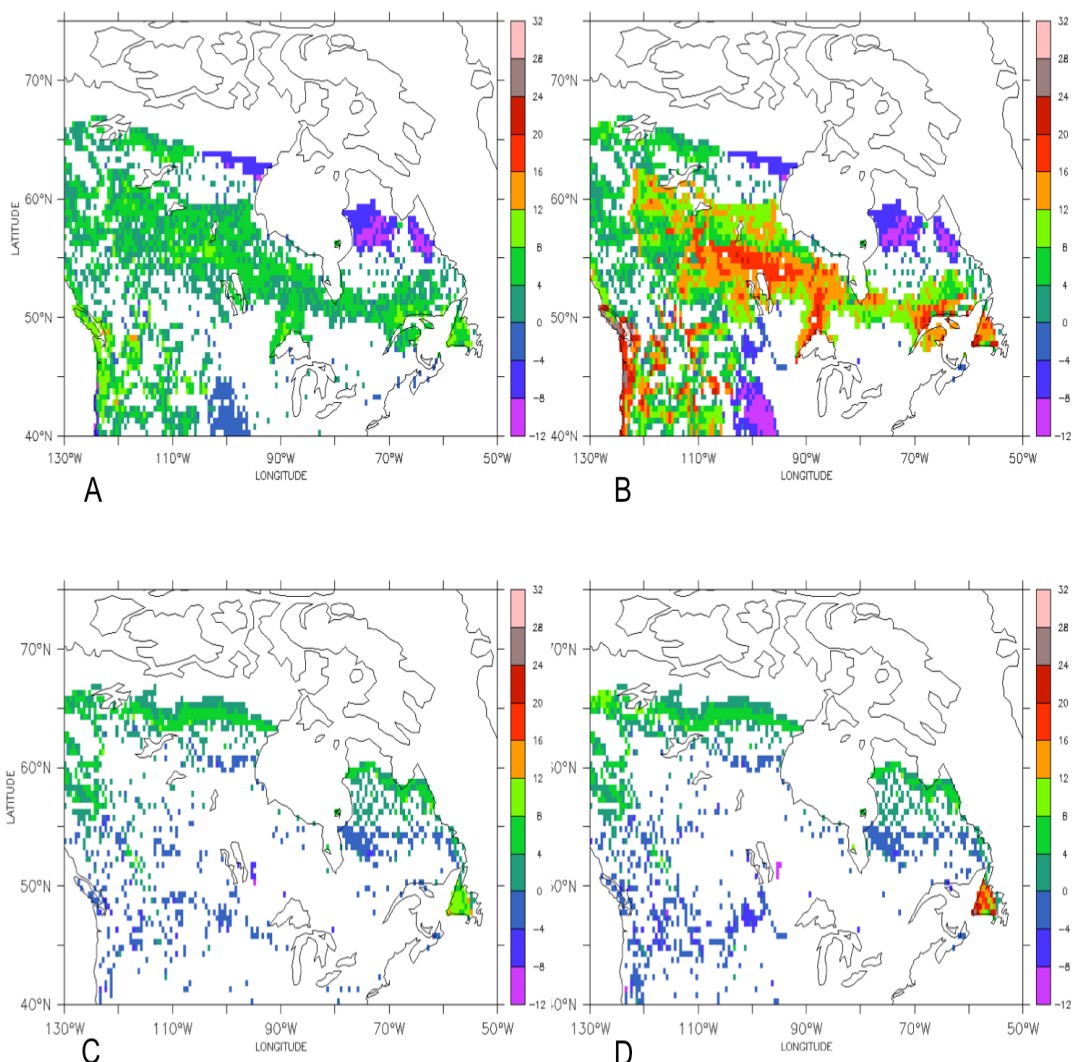

**Figure 9.** Maps after 100 simulation years of (**A**) total carbon difference between the 10 km directed dispersal with standard disturbance scenario and the equilibrium response, and (**B**) the portion from the evergreen PFT. Until the deciduous species migrates, the evergreen PFT stored more carbon than the deciduous PFT. At simulation year 2000, both total carbon difference (**C**) and the proportion from the evergreen PFT (**D**) were similar to the equilibrium response, except for the northern expansion of the forest extent (C and D, green).

## 4. Discussion

This was the first application of the ED migration sub-model that was developed to continue to advance new ways of simulating the transient response of vegetation from individual based dispersal at large domains. Landscape models are adapting the PFT and cohort structure [28,48] used in many DGVMs to increase the size of the domains they simulate, and DGVMs implement various strategies to simulate the transient response [18–21], but are often not individually based or contained between grid cell dispersal. ED, with its pseudo-spatial structure, was then positioned to continue to help advance research in this field, as it is constructed for large domain simulation but is individually based, so when the spatially explicit process of migration was adapted to its pseudo-spatial framework, it offered another model to simulate the transient response of plant migration. Though all scenarios ran at the half-degree resolution that other DGVMs often run at, this occurred from data limitation, not computational limitation. Future climatologies that contain the necessary specific humidity input are often not readily available, so only North America data from the NARCCAP was used. Additionally,

only a portion of that data was used, as transition zones were the focus and ED had previously been validated with remote sensing data on the distribution of deciduous and evergreen PFTs in North America [33]. ED is readily adaptable in both resolution and number of PFTs. For NASA CMS, ED is being run at 90 m resolution for parts of the US [39,41], and NASA GEDI plans to run the contiguous US at 1 km [42], which is on the larger side of most landscape models, but within reason. As ED is a DGVM, specific species are lost, but additional PFTs can be added. The version used here also has three tropical species and two grass species outside of the domain. ED2, a modification of the original ED that is often used in research in smaller domains, used five temperate PFTs for trees in study at Harvard Forest [49] and seven tropical tree PFTs at a Costa Rican site [50], so it can be modified for whatever the research requires. Our findings on the transient response of plant migration in northern North America are consistent with previous studies and present another method for studying the transient response in large domains.

Here, the transient response of plant migration on vegetation and carbon redistribution over a domain where the initial PFT distribution under current climate was verified with remote sensing data [33] was assessed. The transient responses PFT and carbon distribution never matched the equilibrium response for a variety of reasons. The evergreen PFT was outcompeted by the deciduous PFT at many locations as it moved north, but until the deciduous PFT migrated there, the evergreen PFT stored slightly more carbon (Figure 9A,B). Migration also increased the northern forest extent (Figure 5) as sites that were not classified as forest in the equilibrium response exceeded the forest threshold definition when new individuals migrated to those sites (Figure 5). Migration to the predicted equilibrium response only covered the entire domain when the dispersal distance was 10 km per year (Figure 6). So, with new forested areas and the evergreen PFT storing more carbon than the deciduous PFT before it is outcompeted, carbon sequestration potential was almost always higher than the predicted equilibrium response, and the PFT distribution lower from the time it took to migrate and the increased extent of the forest. This study, to our knowledge, is one of the first to examine the transient response of individual-based plant migration with an advanced mechanistic model at continental scales with multiple dispersal rates, dispersal modes, and disturbance rates. Though novel in approach, our results are comparable to previous studies on vegetation and carbon redistribution from climate change.

Modest net changes in total carbon with larger underlying grid changes, presented here, were also found by Schaphoff et al. [51]. Using the LPJ-DGVM with five different general circulation models (GCMs) for a climate change scenario produced an average increase of 7.1% in vegetation carbon across the globe. However, they had boreal forests as a carbon source, whereas we found it to be a temporary sink. This could be a result of the climate change scenario they used, the IS92a. The atmospheric $CO_2$ value used for our research was 575 ppm while they used 703 ppm. Bachelet et al. [52] used an equilibrium model, MAPSS, and a dynamic model, MC1, to simulate changes in potential equilibrium vegetation and carbon distribution in the US, and found that moderate temperature increases produced an increase in carbon with limited redistribution, but higher temperature changes produced widespread redistribution and carbon loss. Solomon and Kirilenko [53] used three climate scenarios to predict future equilibrium distribution carbon with and without migration, and found modest total gains in carbon were the product of larger underlying redistribution of ecosystems. Sitch et al. [16] ran five DGVMs with the A1 scenario and all had the tundra becoming a sink, while their temperate results varied, but overall were also a sink. At the regional scale, Brandt et al. [54] and Jin et al. [27] ran three models, Climate Change Tree Atlas, LANDIS-PRO, and LINKAGES in the central hardwood ecosystem and projected significant changes in species composition with moderate carbon changes. Wang et al. [28] used LANDIS-PRO in the northeastern United States with four climate change scenarios and all found an increase in AGB, with hardwoods replacing conifer species. While not reporting on carbon, Morin [55] looked at 16 North American tree species and their suitable zones, and Iverson [56,57] used various models to predict range shift under climate change, all of which are consistent with our findings. Migration's greatest influence will occur at transition zones; in North America that means

evergreen forests are expected to migrate from the taiga into the tundra [58], and deciduous forests are expected to move northward [59]. Northward migration of boreal species into regions previously classified as tundra is already occurring [60] as remote sensing supports tree line advance [61]. Both of these trends were represented in these previous studies and our research.

As for disturbance, it can both accelerate and impede migration. Disturbance rates control the probability of new species establishment, as some disturbance is needed for new species to enter an ecosystem, but too much prevents establishment [62,63]. The MIGRATE model investigates how available habitat impacts migration rates and shows that increased suitable habitat increases migration rates [64]. FORSKA, a gap model, also showed increased disturbance lead to faster redistribution in the mixed conifer/northern hardwoods zone of northern Europe [65]. Representative of this, with increased disturbance the deciduous PFT migrated and established faster (Figure 7), as it benefited from less competition. However, species are only so resilient to disturbance, so increased disturbance in low biomass areas can impede migration [56]. Our results showed that the northern extent of the evergreen PFT was reduced (Figure 1) as the low growth rates there prevented forest establishment with a higher disturbance rate.

This study has made important advances in using an individual-based mechanistic model to predict the potential transient response of vegetation and carbon to climate change over large domains. Future work should prioritize expansion of the scenarios used and incorporate additional metrics. There are many other studies [17–25] that simulate the transient response in some capacity, but often lack between grid cell dispersal, or are not individually based. The results are supported by previous studies, and offer another method to potentially examine Reid's Paradox of rapid plant migration over large domains, but this was still a simplification of a complex process. Additional PFTs can be added [49,50] depending on the research question. A static dispersal distance was used, but long distance dispersal is governed by the tail of dispersal kernels [10,66], and can be implemented. Only one climate change scenario was used, with a static value of $CO_2$ that is high, but not the highest presented in the SRES. The NARCCAP is producing numerous current and future, and as they are all forced with the A2 scenario, a sensitivity analysis can be performed. The disturbance rate can be PFT specific rather than equal for all types, as climate change is causing increased insect outbreaks that are damaging boreal forests [67], so the disturbance rate could be increased in at-risk areas or for specific PFTs. Fire is also increasing and altering species distribution [68], so ED's fire sub-model could be parameterized and explored. The climatologies were at half-degree but could be downscaled, and if future climatologies for other regions are generated they could be explored. The MsTMIP climate data used for the current climatology was also used in a model intercomparison and demonstrated a wide range in potential changes based on the model [15,69], so another intercomparison could be performed. This research presents a novel method to simulate the transient response of vegetation and carbon to climate change in large domains, and future research should replicate many of the studies that have been conducted at smaller scales on disturbance, dispersal, competition, and landscape characteristics, and be implemented at scales up to global in model intercomparison projects and sensitivity analyses.

## 5. Conclusions

A model experimental design was used to isolate dependencies and explain results of the potential impacts of dispersal distance, dispersal mode, and disturbance rate on the transient response of vegetation and carbon to climate change in northern North America. The major conclusions were: (1) Transient results indicated a temporary increase in carbon sequestration potential relative to equilibrium results, as the photosynthetic response was faster than the competitive response. (2) Potential carbon accumulation was most strongly impacted by dispersal mode, moderately impacted by disturbance rate, and least impacted by dispersal rate. (3) PFT redistribution was most strongly impacted by dispersal rate, moderately impacted by disturbance rate, and least impacted by dispersal mode. These results illustrate the complex transient interactions of biogeography and

biogeochemistry, and support continued research on the impact of plant migration on vegetation and carbon redistribution due to climate change.

**Author Contributions:** Conceptualization, S.A.F. and G.C.H.; Data curation, S.A.F., J.P.F., R.S., and M.Z.; Formal analysis, S.A.F.; Funding acquisition, G.C.H. and R.D.; Investigation, S.A.F.; Methodology, S.A.F.; Project administration, G.C.H.; Resources, G.C.H. and R.D.; Software, S.A.F., J.P.F., R.S., and M.Z.; Supervision, G.C.H., J.P.F., R.D., M.C.H., J.H.S., and G.J.C.; Validation, R.D.; Visualization, S.A.F. and G.C.H.; Writing—original draft, S.A.F., and G.C.H.; Writing—review & editing, R.D., M.C.H., J.H.S., and G.J.C.

**Funding:** This research was funded by the National Aeronautics and Space Administration (NASA) through the Terrestrial Ecology (TE) program (grant number NNX10AO03G), Carbon Monitoring System (CMS) (grant number 80NSSC17K0710 ), and the Interdisciplinary Research in Earth Science (IDS) program (grant numbers NNX10AP11G and 80NSSC17K0348).

**Acknowledgments:** We gratefully acknowledge the support of the NASA Terrestrial Ecology Program, NASA Carbon Monitoring System, NASA Interdisciplinary Research in Earth Science program, and the NASA Earth and Space Science Graduate Fellowship Program.

**Conflicts of Interest:** The authors declare no conflict of interest. The funders had no role in the design of the study; in the collection, analyses, or interpretation of data; in the writing of the manuscript, or in the decision to publish the results.

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
