# Peer review of "Potential Transient Response of Terrestrial Vegetation and Carbon in Northern North America from Climate Change"

_climate, doi:10.3390/cli7090113_

Round 1

Reviewer 1 Report

I thank the authors and the editor for an opportunity to review this manuscript.

This paper presents a modeling exercise where a migration module was added to a dynamic global vegetation model, and the impact of the module is explored through a continental-scale simulation of northern North America. I believe the general concept is interesting and a robust exploration of the concept would have been an original contribution to the literature.

The introduction is poorly written, with many colloquial and imprecise expressions. It falls short of establishing a compelling reason for this particular modeling study. Specifically, it does not explain clearly why past DGVM-based publications fail to capture what the authors want to do. The authors argue for a need for simulating “transient” response of vegetation. Typically, this means simulations that represent changing conditions over time as opposed to an equilibrium condition, and for that type of transient response, there have been many published simulations. I think the authors mean to argue for simulations that explicitly represent plant migration and dispersal mechanisms. Word choice notwithstanding, I don’t think the authors make a persuasive case for why plant migration needs to be simulated at this given scale. I think there may be a case to be made (for this scale) – it’s just that the authors don’t quite establish that case by citing recent DGVM and other terrestrial biome models. Indeed their survey of recent literature is decidedly sparse, and fails to cite many good global and continental scale modeling studies.

The methods fail to adequately describe the migration/dispersal mechanisms that are implemented, including what the various dispersal modes do and how they interact with the ecosystem logic. The authors also do not provide any rationale for the experiment design: the selection of the 18 combinations of dispersal modes and distances. How do these dispersal modes and distances relate to real vegetation dispersal dynamics, and what questions do you hope to answer by choosing these 18 specific combinations? Without that type of rationale, the exercise appears to be a general sensitivity test of a very general concept: can dispersal affect biogeography and carbon storage? The answer the authors arrive at is, unsurprisingly, yes.

The manuscript would be improved by providing (a) a more detailed explanation of the dispersal mechanism, including how it reflects (or doesn’t) real tree migration dynamics; (b) a more careful evaluation of the model calibration, what it captures and fails to capture; (c) what specific ecological questions the exercise aims to answer; and (d) focus the discussion on how what the study demonstrates is relevant to recently published DGVM studies. Many of the references the authors cite are rather old, when many new publications exist from the last 10-15 years. As it stands, the take-home message of the paper is, “can dispersal affect biogeography? Yes” and that is not a compelling message. I think more novel explorations of the dynamics are possible, with stronger take-home messages.

Finally, the manuscript suffers from a somewhat poor quality of writing, with imprecisely worded phrases and sentences, use of colloquial phrases, and some minor grammatical errors. While I did not weight this issue very much, it is a significant hindrance to communicating the results of a study that has the potential to be original with novel results.

Some detailed line-by-line comments:

52-67: Authors cite a 1985 and 1996 paper as motivation for this modeling exercise, and provide a meager review of all the DGVM simulations that have been published in the last 23 years.
83: Please rephrase to say that Landis-II was used to study … (since it’s not a person).
87: “transit” – I think you mean transient.
88: A single 30m x 30m forest was extrapolated? Need to rephrase.
89: “Looking at” is very colloquial.
90: Wall-to-wall should be in double quotes.
91: “inherit” -> inherent
92: I don’t think a persuasive case has been made that there is a computational challenge.
100: “millennium” -> millennia

116-129: How were the dispersal rates and modes, and the disturbance chosen? There needs to be some defensible rationale based on observations/literature values. What does it mean to have a disturbance rate? How exactly does this affect dispersal? What is “even” vs. “directed” dispersal? (I think I can guess what “even” is). You jumped straight from currently climatology to future, without a smooth transition? How were the simulation years (1,000 for current and 36,000 for future) chosen? How does the dispersal mechanism interact with the existing ecophysiology logic?
141: “hourly daily” -> just “hourly”
147: Some characterization of this particular choice of GCM-RCM-scenario combination (CCSM-MM5I-A2) are needed. What type of a future does this projection represent? Is it a particular hot or wet projection, etc.
153: It is unclear how the “equilibrium” case is different from the 18 experiments.

Figure 1: No evaluation of the current equilibrium simulation is given. Does Figure 1-A show a reasonable approximation of the current distribution of the PFTs? Do the carbon stocks compare reasonably well w/ any observation datasets and/or literature values?

Table 2: Presenting the data this way makes it hard to absorb the overall patterns. Please plot it with a chart, strategically choosing the design of the chart to communicate your points.

169-174: That the dispersal mechanism affected migration seems unsurprising, and with warming, it’s also unsurprising that the slower northward migration of a warmer climate-adapted PFT results in lower carbon gain.

225: I suggest rephrasing this to “Temporal patterns of migration,” because the word “transient” is synonymous with “temporal.”

267-278: This is unsurprising, given results of many multi-model comparison studies, which shows that terrestrial models exhibit highly divergent behaviors. I think the discussion should focus more on the impact of these models no having explicit dispersal/migration mechanisms. Also, you discuss modeling exercises from 1997, 2001 and 2006. That seems very dated. There are far more recent terrestrial biome simulations that cover North America.

296: What is “Us again”?

Reviewer 2 Report

The manuscript focuses on assessing the transient response of some plant functional types in northern North America, according to alternative dispersal patterns under future climate paths. The work extends that by Flanagan et al. 2016 (doi:10.3390/cli4010002), which is based on the potential distribution of vegetation over the same area. Input data and model used are similar between the two works, except for the methodology, which incorporates disturbance scenarios, dispersal assumptions, and extended timeframe. I recognize few new information in comparison with the previous work, though they might be worth to publish, after minor revisions. Hereafter some suggestions from my side. - Differentiate the introductory contents from the previous work (Flanagan et al. 2016 (doi:10.3390/cli4010002)). Wording and entire sentences seem copied and pasted. The same applies to part of the discussion section. - A subset of the results presented here is retrieved from Flanagan et al. 2016 (doi:10.3390/cli4010002). I specifically refer to Figure 1 (A-B), Figure 4 (A), and partially the text at lines 162-164. For a sake of transparency, I would strongly recommend to explicitly cite Flanagan et al. 2016 (doi:10.3390/cli4010002) along with the above-mentioned figures, values and text (not limited to). - All the most relevant differences (approach used, model improvement, etc.) with Flanagan et al. 2016 (doi:10.3390/cli4010002) should be highlighted in the discussion section. Hence, it is recommended to expand the sentence at lines 252-255. A bulleted list can be sufficient.

Reviewer 3 Report

Some terms need to be better defined in both the introduction and methods sections.

There is some work to do with the methods section. I am not sure about the value of table one in the text. 

Table two needs some work. The average value of the entire list seems to me not very informative given the various scenarios used. Lines 167 to 174 on page 6 need to be revised in order to clarify the results. 

Figure 2 should be grouped by PFT and Carbon. Discuss from that grouping would be more in line with the overall direction of the text.

ALl figures could use some cleaning up. There are numerous edits that should be carried out in the manuscript as well.

Round 2

Reviewer 1 Report

Thank you for the opportunity to re-review. The manuscript has been improved.

The text is still marred by frequent use of vague and colloquial phrases, such as "we looked at", "we build off of." Many sections read like it was spoken, rather than text composed for precision.

The introduction now ends with three numbered goals, but the entire motivation still feels slim. The only motivation cited is that it's computationally expensive to do, but there is rich literature that demonstrates that dispersal and migration processes are important at the continental scale. (In particular, in paleo-ecological literature).

The experiment design with 18 cases needs a better overview. I suggest using a table to show what the 18 cases are. Currently Figure 2 kind of helps serve that purpose but it’s just a list of all 18 cases. Something more like a matrix would be clearer to the reader. Along those lines, Figure 2 could be made far more impactful by plotting it on 2 or more axes. Just laying it out in a line, left to right, is a lost opportunity.
There is no discussion of what the limited model design (the use of just 2 PFTs and one dispersal distance at a time) has on the conclusions. Real dispersal processes are far more complex. So what does that mean for your results?
There is no discussion of what these results mean for published DGVM simulations in North America and elsewhere. MsTMIP included many transient simulations. What implications do your findings have for those?

60 PFT used w/o being defined.
65 Awkward phrasing
66 “spread” is a bit vague; recommend “dispersal”
69 “looked at” is too colloquial.
79 “build off of” is too colloquial.
351 “Examination of only equilibrium response…” – the alternative to simulating equilibrium response is not to simulate migration/dispersal processes. There are many simulation studies that examined transient response but without migration/dispersal. In other words, “transient” and “dispersal” are not synonymous.
